# Effects of Different Scan Duration on Brain Effective Connectivity among Default Mode Network Nodes

**DOI:** 10.3390/diagnostics12051277

**Published:** 2022-05-20

**Authors:** Nor Shafiza Abdul Wahab, Noorazrul Yahya, Ahmad Nazlim Yusoff, Rozman Zakaria, Jegan Thanabalan, Elza Othman, Soon Bee Hong, Ramesh Kumar Athi Kumar, Hanani Abdul Manan

**Affiliations:** 1Diagnostic Imaging & Radiotherapy Program, Centre for Diagnostic, Therapeutic and Investigative Studies, Faculty of Health Sciences, Universiti Kebangsaan Malaysia, Jalan Raja Muda Abdul Aziz, Kuala Lumpur 50300, Malaysia; shafiza@ppukm.ukm.edu.my (N.S.A.W.); nazlimtrw@ukm.edu.my (A.N.Y.); 2Makmal Pemprosesan Imej Kefungsian (Functional Image Processing Laboratory), Department of Radiology, Universiti Kebangsaan Malaysia Medical Centre, Jalan Yaacob Latif, Bandar Tun Razak, Cheras, Kuala Lumpur 56000, Malaysia; rozman@ppukm.ukm.edu.my; 3Department of Radiology and Intervency, Hospital Pakar Kanak-Kanak (Specialist Children Hospital), Universiti Kebangsaan Malaysia, Jalan Yaacob Latif, Bandar Tun Razak, Cheras, Kuala Lumpur 56000, Malaysia; 4Department of Neurosurgery, University Kebangsaan Malaysia Medical Centre, Jalan Yaacob Latif, Bandar Tun Razak, Cheras, Kuala Lumpur 56000, Malaysia; jegan@ppukm.ukm.edu.my (J.T.); rameshkumar@ppukm.ukm.edu.my (R.K.A.K.); 5School of Medical Imaging, Faculty of Health Sciences, Universiti Sultan Zainal Abidin, Kuala Terengganu 21300, Malaysia; elzaazri@gmail.com; 6Department of Surgery, Pusat Perubatan Universiti Malaya, Lembah Pantai, Kuala Lumpur 59100, Malaysia; bsoon00@yahoo.com

**Keywords:** resting-state functional MRI, scan duration, functional connectivity

## Abstract

Background: Resting-state functional magnetic resonance imaging (rs-fMRI) can evaluate brain functional connectivity without requiring subjects to perform a specific task. This rs-fMRI is very useful in patients with cognitive decline or unable to respond to tasks. However, long scan durations have been suggested to measure connectivity between brain areas to produce more reliable results, which are not clinically optimal. Therefore, this study aims to evaluate a shorter scan duration and compare the scan duration of 10 and 15 min using the rs-fMRI approach. Methods: Twenty-one healthy male and female participants (seventeen right-handed and four left-handed), with ages ranging between 21 and 60 years, were recruited. All participants underwent both 10 and 15 min of rs-fMRI scans. The present study evaluated the default mode network (DMN) areas for both scan durations. The areas involved were the posterior cingulate cortex (PCC), medial prefrontal cortex (mPFC), left inferior parietal cortex (LIPC), and right inferior parietal cortex (RIPC). Fifteen causal models were constructed and inverted using spectral dynamic causal modelling (spDCM). The models were compared using Bayesian Model Selection (BMS) for group studies. Result: The BMS results indicated that the fully connected model was the winning model among 15 competing models for both 10 and 15 min scan durations. However, there was no significant difference in effective connectivity among the regions of interest between the 10 and 15 min scans. Conclusion: Scan duration in the range of 10 to 15 min is sufficient to evaluate the effective connectivity within the DMN region. In frail subjects, a shorter scan duration is more favourable.

## 1. Introduction

The advancement of technology makes non-invasive resting-state functional magnetic resonance imaging (rs-fMRI) one of the tools that can be performed to evaluate the brain’s functional connectivity without requiring subjects to perform any specific task [1,2,3,4]. The blood oxygenation level-dependent (BOLD) technique uses spontaneous fluctuations by detecting changes occurring in the blood flow in the brain [5,6,7]. The magnetic resonance (MR) signal will increase as activity occurs in the brain [8,9]. The advantage of rs-fMRI is that it can identify many intrinsic brain networks, such as the default mode network (DMN), salience network, executive control network (ECN), and language function network, as similarly achieved by task-based fMRI [10,11,12,13,14]. Furthermore, the ability of rs-fMRI to identify the strength of connectivity between networks is being actively studied. One of the most active networks in rs-fMRI is the default mode network (DMN) [15,16,17]. The four central regions of DMN are posterior cingulate cortex (PCC), medial prefrontal cortex (mPFC), left inferior parietal cortex (LIPC), and right inferior parietal cortex (RIPC) [17,18,19,20]. These areas are most likely to exhibit high metabolic activity at rest and during passive sensory processing tasks, while being deactivated during the performance of goal-directed cognitive tasks [21,22,23]. These DMN networks play an essential role in cognitive function, as this is an area where the emotional processes, memory retrieval, and social cognition occur [19,23,24,25,26,27]. It has been suggested that a longer scan duration may give more reliable results in rs-fMRI [28]. The most commonly used scan duration for rs-fMRI should be within 5–7 min because, at this point, the strength of functional connectivity is stable [29]. However, according to Anderson et al., (2011) [30], the best scan duration for rs-fMRI is between 12–20 min to accurately differentiate the functional connectivity of one individual from a group of subjects using an automated machine learning classifier. However, another study by Braun et al., (2012) [31] showed that the scan duration of between 3 to 11 min is enough to evaluate the reliability of rs-fMRI. 

This study evaluates the effect of different scan durations of 10 and 15 min on brain effective connectivity in default mode networks (DMN) among healthy subjects. The results from this study will be implemented in another experiment focusing on brain tumour patients. This pilot study focuses on healthy subjects.

The main highlight of this study is to evaluate the effective connectivity between two different scanning times. Most of the previous studies have no standard protocols in scanning time during rs-fMRI acquisition. This study uses spDCM, a statistical approach that can evaluate how selected brain regions interact with each other. We aim to present a dynamic causal model that could be useful in analysing resting-state studies, especially in the main DMN regions, in different scanning times. For example, a previous study by Friston et al., (2014) [32] used a visual paradigm stimulus to evaluate the effective connectivity in the regions of the early visual cortex (V1), motion-sensitive area (V5), lateral occipital cortex (LOC), posterior parietal cortex (PPC), frontal eye fields (FEF) and prefrontal cortex (PFC). Another study by Han et al., (2020) [33] also used task-based fMRI to determine whether short or long paradigms should be used for brain activation during exposure to odour stimulus. They did not focus on effective connectivity. Therefore, this research is different from other studies since the main objective is to identify the best scanning time in effective connectivity in DMN networks.

## 2. Materials and Methods

### 2.1. Participants in Study

The experimental study was conducted at the Pusat Perubatan Universiti Kebangsaan Malaysia (PPUKM). This work is supported by the Research University Grant Universiti Kebangsaan Malaysia (UKM) GGPM-2017-016. The approval from the ethics community has been obtained (UKM PPI/111/8/JEP-2018-040). The demographic parameters, such as the average age, gender, and handedness ratio of the participants, are tabulated in Table 1. Among the 21 participants, 15 were males, and 17 were right-handed. The average age of the participants was 31.90 ± 1.77 years. All participants had formal education of at least 11 years. All participants passed MRI screening and could cooperate and follow the instructions required for this fMRI study. The MRI examination excluded participants with a history of psychiatric illness, prior psychoactive medication usage, pregnant, clinically unstable, claustrophobic, or contraindicated.

### 2.2. Data Acquisition 

Resting-state fMRI (rs-fMRI) data were collected using the 3T MRI system model Siemens Magnetom Verio at PPUKM. A standard 32-channel head coil was used, along with a restraining foam pad to minimise head motion and with earplugs to minimise scanner noise. The participants were instructed to keep their eyes open and passively focus on a fixation point, “x”, during the entire rs-fMRI scans. They were also instructed to be calm and stay awake during the entire session. The state of consciousness of the participants throughout the scan was verified after the scan was completed. The first experiment started with 10 min of scanning time. A simple question was asked between the scan durations to the subjects, such as whether they were in a comfortable position before starting up the 15 min scanning time. At this point, the subjects were reminded to not fall asleep during the scan. A simple question, such as how they felt when they were inside the MRI and whether they had a headache, was asked verbally after completing the scan. The rs-fMRI data for scan durations of 10 and 15 min were collected during the same experimental session. The first ten scans were dummies and automatically discarded by the BOLD imaging protocol to eliminate the magnetic saturation effect. The pulse sequence used was gradient-echo echo-planar imaging (GRE-EPI) with the following parameters for acquiring functional T2* weighted images: TR/TE = 3000 ms/29 ms, flip angle = 75°, FOV = 240 mm × 240 mm, data matrix = 64 × 64, slice thickness = 3.5 mm, slice gap = 1.05 mm, voxel size = 3.75 × 3.75 × 3.75 mm and number of scans = 200 and 300 for scan durations of 10 min and 15 min, respectively. A series of high-resolution T1-weighted images were also acquired with a volumetric three-dimensional spoiled gradient recall sequence with the following parameters: TR/TE = 2200 ms/3.2 ms, FOV = 256 × 256 mm^2^, matrix size = 256 ×256 and slice thickness = 1 mm.

### 2.3. Pre-Processing

The rs-fMRI data were analysed using MATLAB 7.10.0 (R2018b) (Mathworks Inc., Natick, MA, USA) and statistical parametric mapping (Functional Imaging Laboratory (FIL)) version 12 (SPM12). The functional images in each measurement were randomly checked for any artefacts that could be caused by magnetic field distortion. The images were then entered into a slice timing module for time correction. They were then realigned using the 6-parameter affine transformation in translational (*x*, *y*, and *z*) and rotational (pitch, roll, and yaw) directions to reduce the effects on the overall signal intensity from participant movements. If the magnitude of patient movement was more than 2 mm during the realignment process, the subject was be discarded from the experiment. Figure 1 shows an example of one subject with a magnitude of movement in translation *x*, *y* and *z* and pitch, roll and yaw. This subject only shows a movement below than 1 mm. After realigning the data, a mean image of the series was used to estimate some warping parameters and map these onto a template that already conformed to standard anatomical space (EPI template provided by the Montreal Neurological Institute (MNI)). The normalisation procedure used a 12-parameter affine transformation. The images were then smoothed using an 8-mm full-width at half-maximum (FWHM) Gaussian kernel. 

### 2.4. Selection of Regions of Interest 

The effective connectivity among the regions of interest (ROIs) was examined using dynamic causal modelling (DCM12) (www.fil.ion.ucl.ac.uk/spm/ 1 January 2020). A general linear model (GLM) containing the slice-timed, realigned, normalised, and smoothed images were used for each subject, and a design matrix was constructed and then estimated. The results from the estimated design matrix were used to extract signals from the cerebrospinal fluid (CSF) centred at (0, −40, −5) and white matter (WM) centred at (0, −24, −33) of a 6-mm radius volume of interest (VOI), respectively. The extracted signals from these two regions were then used to construct a new design matrix and then estimated again. The new design matrix, containing signals from the WM and CSF, was then used to extract signals from the 8-mm radius ROIs of the DMN, i.e., mPFC centered at (3, 54, −2), PCC centered at (0, −52, 26), LIPC centered at (−50 −54 36) and RIPC centered at (48 −54 36). The locations of the DMN regions are shown in Figure 2a. Low-frequency fluctuations (LFF), obtained from the DMN regions for scan durations of 10 min and 15 min, are shown in Figure 2b,c.

The waveform is used to evaluate how the selected brain regions, which are PCC, mPFC, LIPC, and RIPC, respond to the BOLD signal imaging in two different scanning times. This waveform shows the observed response and predicted response of the best model for an individual subject. It can be clearly observed that the LFFs can be captured by the DCM model (Di and Biswal 2014) [16]. The time series (right-hand panels) from four regions are the principal eigenvariates of the regions identified, using seed connectivity analyses in individual subjects. These time series were used to invert the DCMs (both spectral and stochastic) (Razi et al., 2015) [34]. A previous study has identified a different coherent between an individual. For example, a study by Biswal et al., (n.d.) [35] was the first to demonstrate the potential of functional connectivity MRI using intrinsic activity correlations. They showed that the BOLD signal time course from a region in the motor cortex was strongly correlated with the contralateral and midline regions within the motor system (Van Dijk et al., 2010) [29]. The coherent fluctuations were readily observed within individual participants, indicating that the method is highly sensitive and raises the possibility of measuring within individual participants. The correlated fluctuations observed by Biswal et al., (n.d.) [35] were manifest while the participants rested passively without any detectable movement, suggesting that the fluctuations were driven by intrinsic activity events constrained by anatomy (Van Dijk et al., 2012) [36].

### 2.5. Modeling of Low-Frequency Fluctuations

Resting-state fMRI is able to capture low-frequency fluctuations (LFF) with frequencies below 0.1 Hz when the brain is at rest (Di and Biswal, 2014) [16]. The GLM, as used by Di and Biswal (2014), was implemented in this study. The GLM contained eight regressors that simulated LFF in 0.01 Hz to 0.08 Hz (Figure 3). Columns 1 to 8, denoted as parameters, represented the Fourier basis set with a 90° phase delay, oscillating at 0.01 Hz (columns 1 and 2), 0.02 Hz (columns 3 and 4), 0.04 Hz (columns 5 and 6), and 0.08 Hz (columns 7 and 8) (Di and Biswal, 2014) [16]. Column 9 represented the effects caused by other factors, such as background noise. The GLM was then estimated. A diagonal *F*-contrast was applied on all the eight regressors to obtain the regions whose variance could be significantly accounted for by the inclusion of the binarised Fourier series regressors (Di and Biswal, 2014) [16]. This was carried out for all the single subjects. A one-sample *t*-test was used to identify group brain functional connectivity due to the LFF, using the *F*-contrast images of all the subjects. The activation areas were then identified using the WFU PickAtlas toolbox (Wake Forest University, Winston-Salem, NC, USA).

### 2.6. Spectral Dynamic Causal Modeling

The time-series signals for each region were then used in specifying and constructing the causal models. Fifteen dynamic causal models containing PCC, mPFC, LIPC, and RIPC as DMN regions were constructed for this study. The models are shown in Figure 4. Models 1 and 4 have two DMN regions with a bidirectional connection. Models 2, 3, 5, and 6 have two DMN regions with a unidirectional connection. Model 7 has four DMN regions that are fully connected. A fully connected model is a model with a bidirectional connection between four selected regions in this study. Model 8 has a unidirectional connection from PCC to mPFC, LIPC, and RIPC. Model 9 has a unidirectional connection from mPFC, LIPC, and RIPC to PCC. Model 10 has bidirectional connections among regions except for LIPC to RIPC and RIPC to LIPC. Model 11 has unidirectional connections from LIPC and RIPC to mPFC, and from LIPC and RIPC to PCC. Model 12 has unidirectional connections from mPFC to LIPC and RIPC, and from PCC to LIPC and RIPC. Model 13 has unidirectional connections from PCC to LIPC and RIPC, and from LIPC and RIPC to mPFC. Model 14 has unidirectional connections from mPFC to LIPC and RIPC, and from LIPC and RIPC to PCC. Model 15 has bidirectional connections among regions except between LIPC and RIPC and between mPFC and PCC.

The causal models were then estimated using spectral DCM (spDCM) to obtain the coupling parameters (effective connectivity) between the regions. Their endogenous fluctuation of activity was recorded and analysed to generate complex cross spectra. The time-invariant covariance of the random fluctuations between regions was then estimated to obtain the cross spectra density, which was then used in estimating the EC between the DMN regions. The EC among the coupled neuronal responses was then estimated using a neuronal plausible power-law model. 

### 2.7. Bayesian Model Selection

The models were then compared by means of Bayesian Model Selection (BMS) for group studies under the fixed effects analyses (FFX) framework (Sharaev et al., 2016; Othman et al., 2019) [23,27] to determine the optimum model that has the best balance between fit (accuracy) and difficulty. The DCM computes posterior probabilities and protects exceedance probabilities at the group level (Rigoux et al., 2014; Stephan et al., 2009; Davey et.al., 2016) [37,38,39]. The protected exceedance probability, which represents the probability of a given model more frequently than the others (above and beyond chance), was our primary measure for model selection (Rigoux et al., 2014) [37]. The strength of effective connectivity and modulatory effects were summarised using random-effects Bayesian Model Averaging (BMA).

### 2.8. Statistical Analysis on Effective Connectivity

A paired *t*-test was performed to evaluate the difference between the 10 and 15 min scan durations in terms of effective connectivity for 21 subjects who underwent both scanning times. The EC in a winning model among four main DMN areas, measured in every single subject, were analysed using the paired t-test in IBM SPSS version 20. A Pearson correlation (r) test was performed to evaluate the significant difference between the 10 min and 15 min scanning times in this study.

### 2.9. Statistical Analysis on Brain Functional Connectivity

A paired *t*-test in MATLAB 7.10.0 (R2018b) (Mathworks Inc., Natick, MA, USA) and statistical parametric mapping (Functional Imaging Laboratory (FIL)) version 12 (SPM12) were performed to evaluate the difference in brain functional connectivity between the 10 and 15 min scan durations. The location of brain functional connectivity was defined using the WFU PickAtlas toolbox (Wake Forest University, Winston-Salem, NC, USA). 

## 3. Results

### 3.1. Brain Functional Connectivity

Figure 5 shows the group result of functional brain connectivity for the scan durations of 10 and 15 min scanning time, thresholded at a corrected significant level (*p*_FWE_) of 0.05 and extent cutoff (*k*_E_) of 20. The result of the FFX analysis for both groups showed the brain functional connectivity in the DMN area of mPFC, LIPC, RIPC, and PCC for 21 participants for both scan durations when the brain was at rest. The locations of the brain functional connectivity were defined from the SPM and WFU Pickatlas (Automatic Anatomical Labelling (AAL)). The functional brain connectivity in 10 min is higher when compared to 15 min. However, all voxels containing a cluster were significantly activated at the peak level. 

### 3.2. Optimum Model

Figure 6 shows the BMS results for the models shown in Figure 4. A BMS analysis was conducted on both scan durations and the results represented the average values for 21 participants. It can be observed that Model 7 was the most optimum model for both 10 and 15 min. The optimum model refers to a model that has the best balance between accuracy and complexity, when compared among 15 models. Model 7 has the highest evidence with a model posterior probability equal to 1. The effective connectivity (in Hz) between the regions obtained from Model 7 is shown in Table 2. 

### 3.3. Effective Connectivity

Figure 7 shows the effective connectivity between the DMN regions for both scan durations. The figure only showed trivial connections with strengths exceeding 0.05 Hz (Razi et al., 2015) [34]. The self-connections were not shown, but the values were all negatives and tabulated in Table 2. There were some inhibit and exhibit connections between the ROI in Model 7. A connection from mPFC → PCC (0.10 Hz) in 10 min showed an inhibit connection, while 15 min showed an exhibit connection of −0.17 Hz. A connection from PCC → mPFC in 10 min showed an exhibit connection of −0.16 Hz. On the other hand, a connection from LIPC → mPFC in 10 min showed an exhibit connection of −0.06 Hz, while an inhibit connection of 0.06 Hz was obtained for 15 min. A connection from mPFC → RIPC for both scan durations showed exhibit connections, whereby 15 min indicated higher strength (−0.10 Hz) compared to 10 min (−0.17 Hz). Similarly, for RIPC → LIPC, which showed an exhibit connection, the10 min scan duration obtained −0.16 Hz and 15 min obtained −0.11 Hz. Significant connections from PCC → LIPC (−0.06 Hz) and LIPC → PCC (0.05 Hz) were identified for 10 min but lower in the 15 min scan duration, where the value of EC was below 0.05 Hz. The connection from PCC → LIPC was exhibit, while LIPC → PCC was inhibit. An exhibit connection was indicated from RIPC → PCC (−0.06 Hz) for the 10 min scan, whereas the 15 min scan duration was inhibit (0.10 Hz). A connection from PCC → RIPC for 15 min was inhibit (0.06 Hz). All the self-connections for the four ROIs in the fully connected models showed exhibit values for both scan durations. For 10 min, the self-connections were PCC (−0.69 Hz), mPFC (−0.74 Hz), LIPC (−0.89 Hz) and RIPC (−0.89 Hz). For 15 min, the self-connections were PCC (−0.87 Hz), mPFC (−0.52 Hz), LIPC (−1.04 Hz), and RIPC (−0.45 Hz).

### 3.4. Correlation Analyses

Data tabulated in Table 3 showed a result of the Pearson correlation (r) test between the 10 min and 15 min scanning times. Data showed a positive correlations between the DMN regions of interest, except for the connection between mPFC → PCC (r = −0.373) and mPFC → RIPC (r = −0.101). It indicates that there is a correlation of effective connectivity in these two different scanning times during the acquisition of rs-fMRI. This research showed a *p*-value of >0.05, except for the connections LIPC → RIPC (0.022), RIPC → PCC (0.026) and RIPC → LIPC (0.015). 

## 4. Discussion

The main objective of this study is to compare the differences in effective connectivity (EC) of different scan durations of 10 and 15 min. Based on the experiment performed, no significant differences between both scan durations were found in terms of effective connectivity. The values of effective connectivity are presented in both scan durations of in this fully connected model that has a bidirectional connection between the regions. Although some of the connections were reported to have a value of below 0.05 Hz, which is known as a non-trivial effect, the winner model had the highest model posterior probability with a value of 1. However, in terms of functional connectivity, it was higher in the short scanning time. Based on the result from the Pearson correlation (r) test, it showed a significant correlation between the DMN regions of interest, except for the connection between mPFC → PCC and mPFC → RIPC, which showed negative correlation values. It indicates that effective connectivity is associated with scanning time during the acquisition of rs-fMRI. 

Most of the previous studies reported that their results are not consistent with other studies. For example, a study by Li et al., (2012) [40] using stochastic DCM showed an influence from PCC to mPFC. However, a study by Di and Biswal (2014) and Jiao et al., (2011) [16,41] showed a causal influence from mPFC to PCC but not vice versa using Granger causality analysis (GCA). A study by Razi et al., (2015) [34] stated that they failed to detect an influence between RIPC to mPFC. The most consistent finding from the previous study is that mPFC is driven by LIPC (Razi et al., 2015; Di and Biswal 2014) [16,34]. Most of the previous studies used a different scanning time for the acquisition of rs-fMRI. For example, a study by Othman et al., (2019) [27] used 7 min, meanwhile Yusoff et al., (2018) used 9 min and 33 s for data acquisition. However, Braun et al., (2012) [31] suggested that a scan duration of 3 to 11 min is adequate to evaluate the reliability of the rs-fMRI. As the time range mentioned above are suggested by the previous research showed that the signal fluctuations of fMRI is already stable to evaluate the intrinsic connection networks (ICN) of the brain (Van Dijk et al., 2010; Fox et al., 2005) [29,42]. Resting-state fMRI depends on the low-frequency signal fluctuations due to functional connectivity within the brain network, which changes slowly (Esposito et al., 2012) [43]. The acquisition of resting-state data for 6 min may provide a brief overview of these slow changes, subsequently changing for another 6 min in the same session. 

Both scan durations in this study have chosen the same model, which is the fully connected model as a winner model through the BMS analysis. However, the value of effective connectivity with a trivial connection of more than 0.05 Hz is more frequently observed in 10 min of scanning time rather than 15 min. The previous study has shown that nontrivial connectivity parameters in total, initial and final models differ only in their magnitude (Sharaev et al., 2016) [23]. These differences neither lead to changes in connectivity patterns in terms of the existing/absence of a particular connection, nor changes in the roles of a specific connection from excitatory to inhibitory (Sharaev et al., 2016) [23]. This means that the winning model is stable at different time frames in terms of its parameters and reflects a relatively stable effective connectivity pattern within the DMN (Sharaev et al., 2016) [23]. This pattern may slightly change in time, but the main driving areas and connections among them remain the same (Sharaev et al., 2016) [23]. Therefore, we can suggest that the subjects were in approximately the same mental state during the first and second phase of the experiment (Sharaev et al., 2016) [23].

A previous study by Han et al., (2020) [33] demonstrated that the short scanning time showed a higher level of brain activation in the insula and orbitofrontal cortex during the short-run paradigm, as compared to the long-run paradigm when experimenting on the effect of odour. A previous study by Han et al., (2020) [33] stated that prolonged odour stimulation (e.g., more than 20 min) might decrease the BOLD signal level. This indicates that the measurements should be short to achieve maximum activation. From this, it could explain why the functional connectivity in the 10 min scanning time is higher than 15 min scanning time.

Other factors contribute to the signal intensity in fMRI data acquisition other than increasing the duration of the scan to evaluate the brain functional connectivity. Regional homogeneity (ReHO) should be evaluated. ReHO is used to characterise the functional homogeneity of resting-state fMRI (rs-fMRI) signals within a small region (Y. Zang et al., 2004; Zuo et al., 2013) [44,45]. In order to have uniform fMRI signal intensity, the pre-processing steps, such as smoothing and normalisation, need to be taken into consideration when processing the data acquisition. Head motion, white matter, and cerebrospinal fluid correction of rs-fMRI time series can significantly improve the reliability of ReHo, spatial smoothing of rs-fMRI time series artificially enhances ReHo intensity and influences its reliability, and a scan duration of 5 min can achieve reliable estimates of ReHo (Zuo et al., 2013) [45]. It has been proven that ReHO measurement in healthy subjects is different from a clinical patient, particularly patients with epilepsy, Parkinson’s disease, Alzheimer’s disease and others (Zuo et al., 2013) [45]. 

It is important to decide on the best scan time in conducting rs-fMRI to receive full cooperation from the sick and healthy patients as the controls. Most of the patients that have undergone a fMRI scan stated that they are more likely to be comforted when they are in the MRI room during the acquisition time (Szameitat, Shen, and Sterr 2009; Hadidi et al., 2014) [46,47]. However, the most uncomfortable feeling that they face is claustrophobia (Hadidi et al., 2014) [47]. A study was carried out previously to evaluate a patient with stroke perception and the healthy controls towards the fMRI scan, indicating that 12.1% of healthy subjects considered a scan duration of between 30 and 60 min as too long, while no patients considered their 30 min scan interval as too long (Szameitat et al., 2009) [46]. According to Szameitat, Shen, and Sterr 2009, most of the participants commented on the fMRI procedure as “noisy”, “tiring”, “slightly claustrophobic”, “nerve-racking”, “head and neck were sore”, “inability to scratch itches”, “back hurt” (two subjects), and “headphones uncomfortable”. 

This study focuses on the effect of scan duration on effective connectivity of 10 and 15 min scan durations. This study does not perform any test-retest reliability between the intersession and intrasession data of individual subjects and grouping analyses. In future studies, it should be directed to analyse the data that focuses on measuring the intraclass correlation coefficient (ICC), regional homogeneity (ReHO) as well as the effect of pre-processing, such as smoothing and normalisation, on a number of voxel activation between different scan times.

## 5. Future Directions

At the moment, we are analysing the data of rs-fMRI between early 5 min and late 5 min in 10 min scanning times. The study’s objective is to compare the effect of effective connectivity between these two phases on the same scanning time in the motor area. 

## 6. Conclusions

In conclusion, the results showed the ability of rs-fMRI to statistically analyse the effective connectivity in DMN regions with different scan durations. It is important to determine the best scan duration, as fMRI experiments may involve brain tumour patients with motor and cognitive function degradation. This experiment suggested that there was no significant difference between the 10 and 15 min scan durations regarding effective connectivity and brain functional connectivity in the DMN regions. Both durations were able to evaluate the functional brain connectivity in the DMN regions. 

## Figures and Tables

**Figure 1 diagnostics-12-01277-f001:**
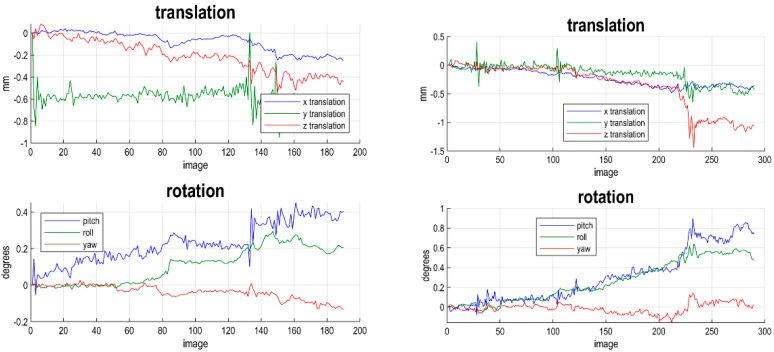
An example of one subject with a magnitude of movement in translation *x*, *y* and *z* and pitch, roll and yaw. This subject only shows a movement below 1 mm.

**Figure 2 diagnostics-12-01277-f002:**
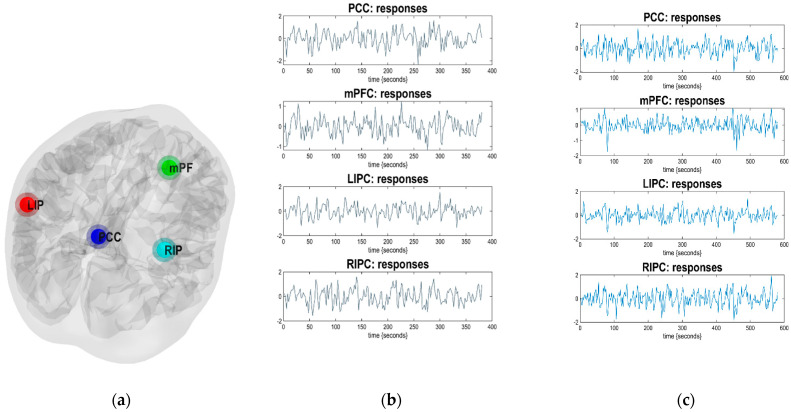
(**a**) The locations of the DMN regions (PCC, mPFC, LIPC and RIPC) on a 3-D brain image and (**b**,**c**) the low-frequency fluctuations (LFF) obtained from the DMN regions for scan durations of 10 min and 15 min. The time series (right-hand panels) from four regions are the principal eigenvariates of regions identified using seed connectivity analyses for a single subject. These time series we used to invert the spectral DCM with the (fully-connected) architecture.

**Figure 3 diagnostics-12-01277-f003:**
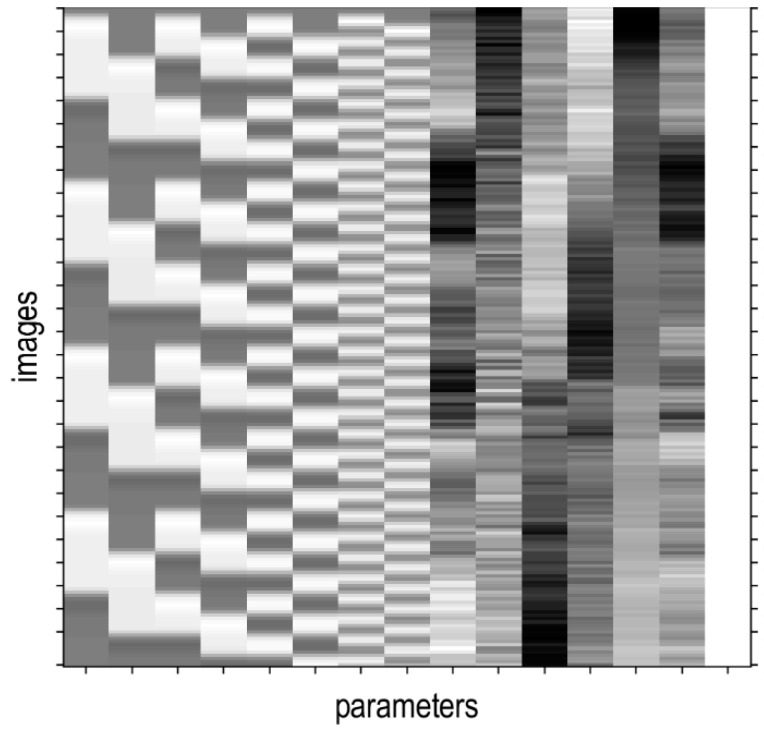
Design matrix for modelling low-frequency fluctuations, containing eight regressors that simulated LFF in 0.01 Hz to 0.08 Hz.

**Figure 4 diagnostics-12-01277-f004:**
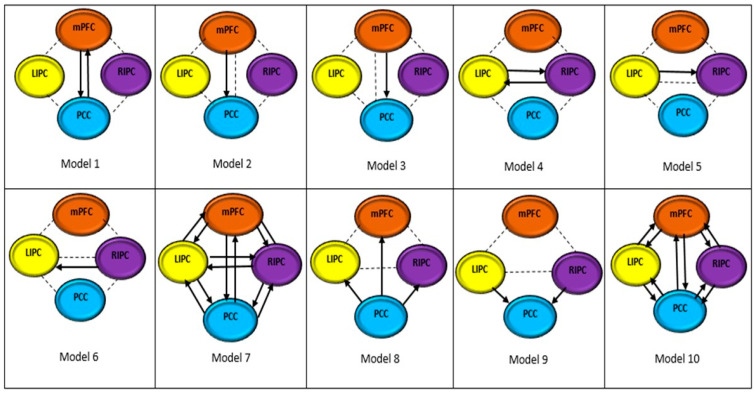
Causal models constructed for comparisons.

**Figure 5 diagnostics-12-01277-f005:**
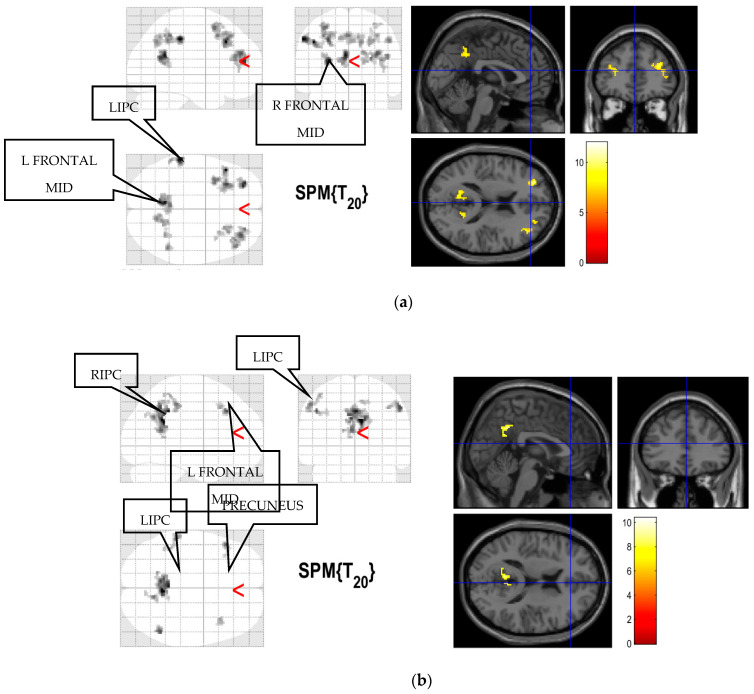
Group result of the brain functional connectivity in DMN obtained from (**a**) 10 and (**b**) 15 min scanning times from 21 healthy subjects (*p*_FWE_ = 0.05, *k*_E_ = 20). A red line arrow head (<) indicate the point of maximum intensity with of the brain activation in each group. Color scales reflect T values of one-sample t test.

**Figure 6 diagnostics-12-01277-f006:**
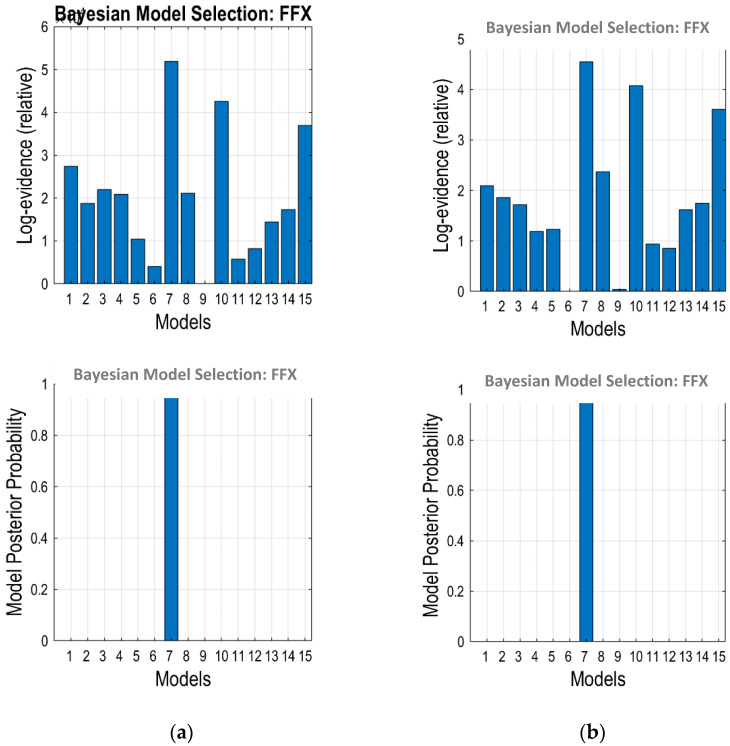
BMS results for (**a**) 10 and (**b**) 15 min scanning times.

**Figure 7 diagnostics-12-01277-f007:**
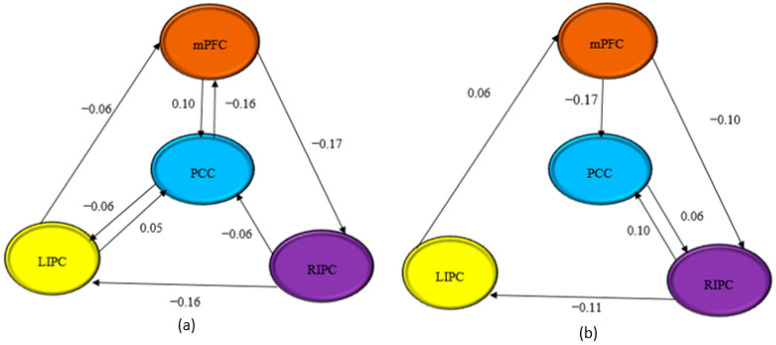
Model 7, the optimum model for connectivity analysis for both scanning times (**a**) 10 and (**b**) 15 min.

**Table 1 diagnostics-12-01277-t001:** Demographic parameters.

Parameters	Data
Gender (M/F)	15/6
Average age/years	31.90 ± 1.77
Age range/years	21 to 60 years old
Handedness (R/L)	17/4

**Table 2 diagnostics-12-01277-t002:** Effective connectivity (in Hz) among DMN nodes for 10 and 15 min scanning times obtained from Model 7.

	10 min	15 min	10 min	15 min	10 min	15 min	10 min	15 min
BMA	From PCC	From PCC	From mPFC	From mPFC	From LIPC	From LIPC	From RIPC	From RIPC
to PCC	−0.6887	−0.8715	0.1023	−0.1741	0.0530	0.0404	−0.0575	0.0980
to mPFC	−0.1626	0.0064	−0.7410	−0.5223	−0.0561	0.0573	0.0301	−0.0099
to LIPC	−0.0554	−0.0384	−0.0427	−0.0248	−0.8941	−1.0364	−0.1556	−0.1129
to RIPC	0.0397	0.0628	−0.1739	−0.0975	−0.0258	−0.0194	−0.8883	−0.4539

**Table 3 diagnostics-12-01277-t003:** Value of *Pearson* correlation (r), R^2^, and *p*-value between 10 min vs. 15 min scanning times.

Effective Connectivity 10 min vs. 15 min Scanning Time	Pearson Correlation (r)	R^2^	*p*-Value
PCC → mPFC	0.418	0.175	0.059
PCC → LIPC	0.204	0.042	0.374
PCC → RIPC	0.350	0.122	0.120
mPFC → PCC	−0.373	0.139	0.096
mPFC →LIPC	0.303	0.092	0.182
mPFC → RIPC	−0.101	0.010	0.064
LIPC → PCC	0.420	0.177	0.058
LIPC → mPFC	0.348	0.121	0.123
LIPC → RIPC	0.495	0.245	0.022
RIPC → PCC	0.486	0.236	0.026
RIPC → mPFC	0.063	0.004	0.788
RIPC → LIPC	0.524	0.275	0.015

## Data Availability

Not applicable.

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
