# Peer review of "Effects of Different Scan Duration on Brain Effective Connectivity among Default Mode Network Nodes"

_diagnostics, 2022, doi:10.3390/diagnostics12051277_

Round 1

Reviewer 1 Report

This is a straight forward paper.

I noticed that GLM was used as part of the experiment but the coefficient of correlation (r2) or correlation coefficient (r) was not shown. It my halp the paper as it will indicate the goodness of fit

Author Response

Dear Editor,

We would like to thank you for the opportunity to revise and resubmit our manuscript entitled " Effects of different scan duration on brain effective connectivity among default mode network nodes" for publication in DIAGNOSTICS. We also appreciate the reviewers for taking some of their precious time and effort to review and assess our manuscript. We are grateful to the reviewers for the constructive comments and suggestions, which will provide some valuable feedback for our manuscript.

We have incorporated most of the comments and suggestions made by the reviewers into our revised manuscript. We have included the reviewers' comments and suggestions below, as well as our responses. The corresponding changes were highlighted in our revised manuscript.

Thank you again for considering our revised manuscript.

Sincerely,

Nor Shafiza Abdul Wahab

Master candidate

Reviewer 1

I noticed that GLM was used as part of the experiment but the coefficients of correlation (r2) or coefficients correlation (r) was not shown. It may help the paper as it will indicate the goodness of fit.

Thank you for the comment from the reviewer. The result has been added into the manuscript. Pearson correlation test was performed to correlate a significant difference in effective connectivity between 10 minutes and 15 minutes scanning times. The result are shown in table 3 in the manuscript.

Pearson correlation (r) test between 10 minutes and 15 minutes scanning times showed positive correlations between the DMN regions of interest except for the connection between mPFCà PCC (r = -0.373) and mPFC àRIPC (r = -0.101). It’s indicated that there is a correlation of effective connectivity in these two different scanning times during acquisition rs-fMRI. This research showed a p-value are > 0.05 except for the connection LIPC àRIPC (0.022), RIPC à PCC (0.026) and RIPC à LIPC (0.015). 

The value of effective connectivity are presented in both scan durations of  10 and 15 minutes scanning time in this fully connected model that has a bidirectional connection between the regions. Although some of the connections are reported to have a value of below 0.05 Hz which is known as a non-trivial effect, the winner model has the highest model posterior probability with a value of 1.

 Most of the previous studies reported that their result are not consistent with other studies. For example, a study by Li et al. (2012) using stochastic DCM showed an influence from PCC to mPFC. However, a study by Di & Biswal (2014) and Jiao et al. (2011) showed a causal influence from mPFC to PCC but not vice versa using Granger Causality Analysis (GCA). A study by Razi et al. (2015) stated that they failed to detect an influence between RIPC to mPFC. The most consistent finding from the previous study is that mPFC is driven by LIPC (Razi et al. 2015; Di & Biswal 2014).

Most of the previous studies used a different scanning time for the acquisition of rs-fMRI. For example, a study by Othman et al. (2019) used 7 minutes meanwhile Yusoff et al. (2018) used 9 minutes and 33 seconds for data acquisition. However, according to Braun et al. (2012) suggested that a scan duration of 3 to 11 minutes is adequate to evaluate the reliability of the rs- fMRI. In that time range, the signal fluctuations of fMRI to evaluate the intrinsic connection network (ICN) of the brains is already stable (Fox et al. 2005; Van Dijk et al. 2010). Resting-state fMRI depends on the low-frequency signal fluctuations due to functional connectivity within the brain network, which changes slowly (Esposito et al. 2012). The acquisition of resting-state data for 6 minutes may provide a brief overview of these slow changes, subsequently changing for another 6 minutes in the same session.

* Refer to table 3: Value of Pearson correlation(r), R2, and p-value between 10 minutes vs 15 minutes scanning times.

*Refer to lines 313- 318

*Refer to lines 327-354

Reviewer 2

  1. The optimal duration of rs-fMRI has been exclusively studied as authors pointed out. It would not be so clear what problems that the authors found in the previous studies and what kind of information that they wanted to add.

The main highlight of this study is to evaluate the effective connectivity between two different scanning times to be applied to the brain tumour patient. There are no standard protocols in scanning time during rs-fMRI acquisition in most of the previous study. This study used a spDCM that is known as a statistical approach that can be used to evaluate how selected brain regions interact with each other. We hope to have introduced a dynamic causal model that could be useful in analyzing resting-state studies, especially in the main DMN regions that is more specifically on PCC, mPFC, LIPC, and RIPC using rs-fMRI in such a different scanning time. For example, a previous study by Friston et al. (2014) used a visual paradigm stimulus to evaluate the effective connectivity at the regions of early visual cortex (V1), motion sensitive area (V5), lateral occipital cortex (LOC), posterior parietal cortex (PPC), frontal eye fields (FEF) and prefrontal cortex (PFC). Another study by Han et al. (2020) also used task-based fMRI in determining either a short or long paradigm should be used for brain activation during the acquisition for odour stimulus and they did not focus on the effective connectivity. So, this research is different from other studies since the main objective is to identify the best scanning time in effective connectivity in the DMN network.

*Refer to lines 79-92

  1. It would be interesting for the readers why the connectivity was not so different, because the raw waveforms look very different.

Thank you for the comment from the reviewer. Figure 2 in the text is used to describe the locations of the DMN regions on a 3-D brain image. The waveform is used to evaluate how the selected brain regions which are PCC, mPFC, LIPC, and RIPC respond to the BOLD signal imaging in two different scanning times. That waveform shows the observed response and predicted response of the best model for an individual subject. It can be seen clearly that the LFFs can be captured by the DCM model (Di & Biswal 2014). The time series (right-hand panels) from four regions are the principal eigenvariates of regions identified using seed connectivity analyses in individual subject. These time series we used to invert the DCMs (both spectral and stochastic) (Razi et al. 2015). Previous study has identified a different coherent between an individual. For example study by Biswal et al. (n.d.) was the first to demonstrate the potential of functional connectivity MRI using intrinsic activity correlations. They showed that the BOLD signal time course from a region in the motor cortex was strongly correlated with the contralateral and midline regions within the motor system (Van Dijk et al. 2010). The coherent fluctuations were readily observed within individual participants, indicating that the method is highly sensitive and raising the possibility of measuring within individual participants. The correlated fluctuations observed by Biswal et al. (n.d.) were manifest while participants rested passively without any detectable movement, suggesting the fluctuations were driven by intrinsic activity events constrained by anatomy (Van Dijk et al. 2012).

Both scan durations in this study have chosen the same model which is the fully connected model as a winner model through the BMS analysis. However, the value of effective connectivity that has a trivial connection that is more than 0.05 Hz are more to be seen in 10 minutes scanning time rather than 15 minutes. Previous study has shown that nontrivial connectivity parameters in total, initial and final models differ only in their magnitude (Sharaev et al. 2016). These differences neither lead to changes in connectivity patterns in terms of existing/absence of a particular connection, nor to changes in roles of a particular connection from being excitatory to inhibitory (Sharaev et al. 2016). This means that the winning model is stable at different time frames in terms of its parameters and reflects a relatively stable effective connectivity pattern within the DMN (Sharaev et al. 2016). This pattern may slightly change in time, but the main driving areas and connections among them remain the same (Sharaev et al. 2016). So, we can suggest that the subjects were in approximately the same mental state during the first and second phase of the experiment (Sharaev et al. 2016).

     *Refer to lines 163-180

*Refer to lines 356-369

  1. It would also be of interest to know how the authors treated motion effects in the processing.

One of the most difficult part to encounter when analyzing the data of rs-fMRI is the motion artifacts especially due to the head motion that can cause artifacts and inconsistency of the data. In this experiment, during the pre-processing step, the first 10 volumes of each data run were discarded to allow for T1-equilibration effects. During pre-processing data, all the data will undergo a realignment process. Other than that, during realignment process, if the magnitude of patient movement is more than 2 mm, the subject will be discarded from the experiment. Besides that, six head motion parameters were also added into the model to remove potential confounding variances caused by head motion. The GLM model also includes an implicit high pass filter of 1/100 Hz, to remove ultraslow fluctuations that were due to scanner drift (Di & Biswal 2014).

Example of one subject with a magnitude of movement in translation x, y, and z and pitch, roll, and yaw. This subject only shows a movement below than 2 mm.

*Refer to lines 139-140

*Refer figure 1

  1. It would be useful to describe how the authors think the optimal duration should be determined. Comparing 3 and 5 minutes in 21 subjects next?

Thank you for the suggestion from the reviewer. This suggestion has been added on the sub-topic future direction. At the moment, we are analyzing the data of rs-fMRI between early 5 minutes and late 5 minutes in 10 minutes scanning time. The study's objective is to compare the effects in effective connectivity between these two different phase on the same scanning time. The selection regions of interest are in the motor area.  Hopefully, based on the current study, it will benefit the readers in the future, especially those interested in this field area.

*Refer to lines 412- 414

Reviewer 2 Report

This study compares the effective connectivity obtained from resting-state (rs) fMRI of 10 and 15 minutes recordings in 21 healthy subjects. The authors calculated the connectivity measures by applying a model representing the default mode network. The results did not show statistical difference between 10 and 15 minutes recording. The whole study is well-described and easy to understand.

Here are a few comments.

  1. The optimal duration of rs-fMRI scans has been exclusively studied as authors pointed out. It would not be so clear what problems the authors found in the previous studies and what kind of information they wanted to add.
  2. It would be interesting for the readers to know why the connectivity was not so different, because the raw waveforms look very different.
  3. It would also be of interest to know how the authors treated motion effects in the processing.
  4. It would be useful to describe how the authors think the optimal duration should be determined. Comparing 3 and 5minutes in 21 subjects next?

Author Response

Dear Editor,

We would like to thank you for the opportunity to revise and resubmit our manuscript entitled " Effects of different scan duration on brain effective connectivity among default mode network nodes" for publication in DIAGNOSTICS. We also appreciate the reviewers for taking some of their precious time and effort to review and assess our manuscript. We are grateful to the reviewers for the constructive comments and suggestions, which will provide some valuable feedback for our manuscript.

We have incorporated most of the comments and suggestions made by the reviewers into our revised manuscript. We have included the reviewers' comments and suggestions below, as well as our responses. The corresponding changes were highlighted in our revised manuscript.

Thank you again for considering our revised manuscript.

Sincerely,

Nor Shafiza Abdul Wahab

Master candidate

Reviewer 1

I noticed that GLM was used as part of the experiment but the coefficients of correlation (r2) or coefficients correlation (r) was not shown. It may help the paper as it will indicate the goodness of fit.

Thank you for the comment from the reviewer. The result has been added into the manuscript. Pearson correlation test was performed to correlate a significant difference in effective connectivity between 10 minutes and 15 minutes scanning times. The result are shown in table 3 in the manuscript.

Pearson correlation (r) test between 10 minutes and 15 minutes scanning times showed positive correlations between the DMN regions of interest except for the connection between mPFCà PCC (r = -0.373) and mPFC àRIPC (r = -0.101). It’s indicated that there is a correlation of effective connectivity in these two different scanning times during acquisition rs-fMRI. This research showed a p-value are > 0.05 except for the connection LIPC àRIPC (0.022), RIPC à PCC (0.026) and RIPC à LIPC (0.015). 

The value of effective connectivity are presented in both scan durations of  10 and 15 minutes scanning time in this fully connected model that has a bidirectional connection between the regions. Although some of the connections are reported to have a value of below 0.05 Hz which is known as a non-trivial effect, the winner model has the highest model posterior probability with a value of 1.

 Most of the previous studies reported that their result are not consistent with other studies. For example, a study by Li et al. (2012) using stochastic DCM showed an influence from PCC to mPFC. However, a study by Di & Biswal (2014) and Jiao et al. (2011) showed a causal influence from mPFC to PCC but not vice versa using Granger Causality Analysis (GCA). A study by Razi et al. (2015) stated that they failed to detect an influence between RIPC to mPFC. The most consistent finding from the previous study is that mPFC is driven by LIPC (Razi et al. 2015; Di & Biswal 2014).

Most of the previous studies used a different scanning time for the acquisition of rs-fMRI. For example, a study by Othman et al. (2019) used 7 minutes meanwhile Yusoff et al. (2018) used 9 minutes and 33 seconds for data acquisition. However, according to Braun et al. (2012) suggested that a scan duration of 3 to 11 minutes is adequate to evaluate the reliability of the rs- fMRI. In that time range, the signal fluctuations of fMRI to evaluate the intrinsic connection network (ICN) of the brains is already stable (Fox et al. 2005; Van Dijk et al. 2010). Resting-state fMRI depends on the low-frequency signal fluctuations due to functional connectivity within the brain network, which changes slowly (Esposito et al. 2012). The acquisition of resting-state data for 6 minutes may provide a brief overview of these slow changes, subsequently changing for another 6 minutes in the same session.

* Refer to table 3: Value of Pearson correlation(r), R2, and p-value between 10 minutes vs 15 minutes scanning times.

*Refer to lines 313- 318

*Refer to lines 327-354

Reviewer 2

  1. The optimal duration of rs-fMRI has been exclusively studied as authors pointed out. It would not be so clear what problems that the authors found in the previous studies and what kind of information that they wanted to add.

The main highlight of this study is to evaluate the effective connectivity between two different scanning times to be applied to the brain tumour patient. There are no standard protocols in scanning time during rs-fMRI acquisition in most of the previous study. This study used a spDCM that is known as a statistical approach that can be used to evaluate how selected brain regions interact with each other. We hope to have introduced a dynamic causal model that could be useful in analyzing resting-state studies, especially in the main DMN regions that is more specifically on PCC, mPFC, LIPC, and RIPC using rs-fMRI in such a different scanning time. For example, a previous study by Friston et al. (2014) used a visual paradigm stimulus to evaluate the effective connectivity at the regions of early visual cortex (V1), motion sensitive area (V5), lateral occipital cortex (LOC), posterior parietal cortex (PPC), frontal eye fields (FEF) and prefrontal cortex (PFC). Another study by Han et al. (2020) also used task-based fMRI in determining either a short or long paradigm should be used for brain activation during the acquisition for odour stimulus and they did not focus on the effective connectivity. So, this research is different from other studies since the main objective is to identify the best scanning time in effective connectivity in the DMN network.

*Refer to lines 79-92

  1. It would be interesting for the readers why the connectivity was not so different, because the raw waveforms look very different.

Thank you for the comment from the reviewer. Figure 2 in the text is used to describe the locations of the DMN regions on a 3-D brain image. The waveform is used to evaluate how the selected brain regions which are PCC, mPFC, LIPC, and RIPC respond to the BOLD signal imaging in two different scanning times. That waveform shows the observed response and predicted response of the best model for an individual subject. It can be seen clearly that the LFFs can be captured by the DCM model (Di & Biswal 2014). The time series (right-hand panels) from four regions are the principal eigenvariates of regions identified using seed connectivity analyses in individual subject. These time series we used to invert the DCMs (both spectral and stochastic) (Razi et al. 2015). Previous study has identified a different coherent between an individual. For example study by Biswal et al. (n.d.) was the first to demonstrate the potential of functional connectivity MRI using intrinsic activity correlations. They showed that the BOLD signal time course from a region in the motor cortex was strongly correlated with the contralateral and midline regions within the motor system (Van Dijk et al. 2010). The coherent fluctuations were readily observed within individual participants, indicating that the method is highly sensitive and raising the possibility of measuring within individual participants. The correlated fluctuations observed by Biswal et al. (n.d.) were manifest while participants rested passively without any detectable movement, suggesting the fluctuations were driven by intrinsic activity events constrained by anatomy (Van Dijk et al. 2012).

Both scan durations in this study have chosen the same model which is the fully connected model as a winner model through the BMS analysis. However, the value of effective connectivity that has a trivial connection that is more than 0.05 Hz are more to be seen in 10 minutes scanning time rather than 15 minutes. Previous study has shown that nontrivial connectivity parameters in total, initial and final models differ only in their magnitude (Sharaev et al. 2016). These differences neither lead to changes in connectivity patterns in terms of existing/absence of a particular connection, nor to changes in roles of a particular connection from being excitatory to inhibitory (Sharaev et al. 2016). This means that the winning model is stable at different time frames in terms of its parameters and reflects a relatively stable effective connectivity pattern within the DMN (Sharaev et al. 2016). This pattern may slightly change in time, but the main driving areas and connections among them remain the same (Sharaev et al. 2016). So, we can suggest that the subjects were in approximately the same mental state during the first and second phase of the experiment (Sharaev et al. 2016).

     *Refer to lines 163-180

*Refer to lines 356-369

  1. It would also be of interest to know how the authors treated motion effects in the processing.

One of the most difficult part to encounter when analyzing the data of rs-fMRI is the motion artifacts especially due to the head motion that can cause artifacts and inconsistency of the data. In this experiment, during the pre-processing step, the first 10 volumes of each data run were discarded to allow for T1-equilibration effects. During pre-processing data, all the data will undergo a realignment process. Other than that, during realignment process, if the magnitude of patient movement is more than 2 mm, the subject will be discarded from the experiment. Besides that, six head motion parameters were also added into the model to remove potential confounding variances caused by head motion. The GLM model also includes an implicit high pass filter of 1/100 Hz, to remove ultraslow fluctuations that were due to scanner drift (Di & Biswal 2014).

Example of one subject with a magnitude of movement in translation x, y, and z and pitch, roll, and yaw. This subject only shows a movement below than 2 mm.

*Refer to lines 139-140

*Refer figure 1

  1. It would be useful to describe how the authors think the optimal duration should be determined. Comparing 3 and 5 minutes in 21 subjects next?

Thank you for the suggestion from the reviewer. This suggestion has been added on the sub-topic future direction. At the moment, we are analyzing the data of rs-fMRI between early 5 minutes and late 5 minutes in 10 minutes scanning time. The study's objective is to compare the effects in effective connectivity between these two different phase on the same scanning time. The selection regions of interest are in the motor area.  Hopefully, based on the current study, it will benefit the readers in the future, especially those interested in this field area.

*Refer to lines 412- 414

This manuscript is a resubmission of an earlier submission. The following is a list of the peer review reports and author responses from that submission.